# Spontaneous Ultraslow Na^+^ Fluctuations in the Neonatal Mouse Brain

**DOI:** 10.3390/cells9010102

**Published:** 2019-12-31

**Authors:** Lisa Felix, Daniel Ziemens, Gerald Seifert, Christine R. Rose

**Affiliations:** 1Institute of Neurobiology, Faculty of Mathematics and Natural Sciences, Heinrich Heine University Duesseldorf, 40225 Duesseldorf, Germany; Lisa.Felix@hhu.de (L.F.); Daniel.Ziemens@hhu.de (D.Z.); 2Institute of Cellular Neurosciences, Medical Faculty, University of Bonn, D-53105 Bonn, Germany; Gerald.Seifert@ukbonn.de

**Keywords:** astrocytes, postnatal development, hippocampus, GABA, neuron-glia interaction

## Abstract

In the neonate forebrain, network formation is driven by the spontaneous synchronized activity of pyramidal cells and interneurons, consisting of bursts of electrical activity and intracellular Ca^2+^ oscillations. By employing ratiometric Na^+^ imaging in tissue slices obtained from animals at postnatal day 2–4 (P2–4), we found that 20% of pyramidal neurons and 44% of astrocytes in neonatal mouse hippocampus also exhibit transient fluctuations in intracellular Na^+^. These occurred at very low frequencies (~2/h), were exceptionally long (~8 min), and strongly declined after the first postnatal week. Similar Na^+^ fluctuations were also observed in the neonate neocortex. In the hippocampus, Na^+^ elevations in both cell types were diminished when blocking action potential generation with tetrodotoxin. Neuronal Na^+^ fluctuations were significantly reduced by bicuculline, suggesting the involvement of GABA_A_-receptors in their generation. Astrocytic signals, by contrast, were neither blocked by inhibition of receptors and/or transporters for different transmitters including GABA and glutamate, nor of various Na^+^-dependent transporters or Na^+^-permeable channels. In summary, our results demonstrate for the first time that neonatal astrocytes and neurons display spontaneous ultraslow Na^+^ fluctuations. While neuronal Na^+^ signals apparently largely rely on suprathreshold GABAergic excitation, astrocytic Na^+^ signals, albeit being dependent on neuronal action potentials, appear to have a separate trigger and mechanism, the source of which remains unclear at present.

## 1. Introduction

The first week after birth constitutes a time of dynamic rearrangement within the mammalian CNS. In the mouse brain, this critical period sees shifts in cellular morphology, connectivity, and protein expression profiles as neurons and glia transform into their mature forms. Neurogenesis in rodents is mostly completed before birth and the neonatal period is characterized by synchronized, universal activity—which acts to guide the maturation of individual cells and their integration into the complex networks critical to the function of mature tissue [1,2,3]. This activity includes early network oscillations (ENOs) in intracellular calcium (Ca^2+^) [4] and later, the related giant depolarizing potentials (GDPs) of neurons [5]. Excitatory activity in neonates’ hippocampus and neocortex has mainly been attributed to GABAergic transmission from interneurons, with additionally modulatory roles for glutamate [6]. Glutamatergic excitation takes over at the end of the first postnatal week, and major synaptogenesis and synapse maturation start to surge in the 2nd and 3rd week after birth [7,8].

Gliogenesis is ongoing during this early stage [9,10] and newly formed astrocytes differ from their mature counterparts in several ways. The density of astrocytes strongly increases (e.g., [11]), and gap junctional coupling between astrocytes develops during the first postnatal week [12]. Moreover, astrocytes undergo significant changes in the electrophysiological properties and alter their expression profile and repertoire of ion channels [13,14,15,16,17]. This process of differentiation and maturation during the first postnatal week produces a shift in GABA-induced intracellular Ca^2+^ signaling [18] and of the subcellular expression profile of Na^+^-dependent GABA transporters [19]. For glutamate transport, the maturing cells initially express primarily glutamate-aspartate-transporter (GLAST) with glutamate transporter 1 (GLT-1) being barely detectable in neonates. GLT-1 levels then strongly increase after the first postnatal week and represents the major glutamate transporter in juveniles and adults [11,20].

While astrocytes are electrically non-excitable cells, a typical and prominent pattern of activity found in mature cells is intracellular Ca^2+^ signaling [21,22]. Pathways in the immature brain are less well reported on, but also involve the generation of spontaneous and evoked Ca^2+^ transients [23,24]. As mentioned above, immature astrocytes already express—albeit at low levels—transporters for GABA and glutamate and both are coupled to the influx of Na^+^ [25,26,27]. Notably, our recent work has demonstrated that disinhibition induced by removal of external Mg^2+^ and application of the GABA_A_-receptor antagonist bicuculline results in network Na^+^ oscillations in the neonate hippocampus, encompassing both neurons as well as astrocytes [28]. Evoked network Na^+^ oscillations required neuronal action potential generation as well as functional glutamate transport [28], demonstrating that neuronal activity and release of glutamate in the neonate brain can result in well-detectable elevations in both neuronal and astrocyte Na^+^, even at this early stage of development.

In the present work, we studied if neurons and astrocytes in the neonate brain might also undergo spontaneous oscillations in intracellular Na^+^. As Na^+^ provides the driving force for a multitude of cellular processes, such Na^+^ oscillations might play an important role in early network formation. To test this hypothesis, we used sodium-binding benzofuran isophthalate-acetoxymethyl ester (SBFI-AM), a ratiometric Na^+^ indicator to monitor somatic Na^+^ changes in astrocytes and neurons in acutely isolated tissue slices of the neonatal mouse hippocampus (P2–4). We found that 20% of neurons and 44% of astrocytes exhibit spontaneous, ultraslow Na^+^ fluctuations that are largely restricted to the first postnatal week and greatly reduced in animals older than P14.

## 2. Materials and Methods

### 2.1. Preparation of Tissue Slices

This study was carried out in accordance with the institutional guidelines of the Heinrich Heine University Düsseldorf, as well as the European Community Council Directive (2010/63/EU). All experiments were communicated to and approved by the animal welfare office of the animal care and use facility of the Heinrich Heine University Düsseldorf (institutional act number: O52/05). In accordance with the German animal welfare act (Articles 4 and 7), no formal additional approval for the post-mortem removal of brain tissue was necessary. In accordance with the recommendations of the European Commission [29], animals up to 10 days old were killed by decapitation, while older mice were anaesthetized with CO_2_ before the animals were quickly decapitated.

Acute slices of mouse hippocampus or neocortex (*mus musculus*, Balb/C; both sexes) of 250 µm thickness were generated using the methods previously published [30]. In addition, transgenic mice were used which displayed unrestricted deletion of connexin30, as well as Cre-recombinase-dependent deletion of connexin43 [31]. Cre-recombinase negative mice were used in control experiments for this section. For animals from postnatal days 10–17 (P10–P17), the dissection and slicing procedure were performed in cooled, modified artificial cerebrospinal fluid (mACSF), which contained a reduced CaCl_2_ concentration (0.5 mM) and an increased MgCl_2_ concentration (6 mM) as compared to standard ACSF (see below) to dampen synaptic release of transmitters and resulting excitotoxic effects during the cutting of the tissue. In addition, the mACSF contained (in mM): 125 NaCl, 2.5 KCl, 1.25 NaH_2_PO_4_, 26 NaHCO_3_, and 20 glucose. This solution had an osmolality of 308–312 mOsm/L and was bubbled with 95% O_2_/5% CO_2_ to achieve a pH of ~7.4. For animals younger than P10 (which are less prone to excitotoxicity), standard ASCF (composition like mACSF apart from 2 mM CaCl_2_ and 1 mM MgCl_2_ in the solution) was used during all steps. For animals older than P18, another modified ACSF was used with (in mM); 60 NaCl, 2.5 KCl, 0.5 CaCl_2_, 7 MgCl_2_, 1.25 NaH_2_PO_4_, 1.3 ascorbic acid, 3 C_3_H_3_NaO_3_, 26 NaHCO_3_, 10 glucose, and 105 sucrose (sucrose added for neuroprotection; recipe modified from [32]). Directly after cutting, slices were incubated at 34 °C for 20 min with 0.5–1 µM sulforhodamine 101 (SR101) in order to stain astrocytes [17]. Following this, slices were transferred to standard ACSF for a further 10 min without SR101.

After this, slices were kept in ACSF at room temperature (20–22 °C) until used in experiments. Throughout experiments, slices were continuously perfused with standard ACSF at room temperature (unless otherwise specified). For experiments performed at near-physiological temperature (34 °C), an ACSF composed of (in mM): 131 NaCl, 2.5 KCl, 0.5 CaCl_2_, 1.3 MgSO_4_7H_2_0, 1.25 NaH_2_PO_4_, 21 NaHCO_3_, and 10 glucose and was used. Pharmacological substances were diluted in ACSF and bath applied to slices via the perfusion system for 15 min before the beginning of, and throughout the measurements. The different substances, the concentrations employed and manufacturers of the chemicals are detailed in Table 1.

### 2.2. Fluorescence-Based Ion Imaging

Slices were loaded via bolus injection (using a picospritzer 3, Parker, Cologne, Germany) with either SBFI-AM (sodium-binding benzofuran isophthalate-acetoxymethyl ester; Invitrogen, Schwerte, Germany) to measure [Na^+^]_i_, BCECF-AM (2’,7’-Bis-(2-Carboxyethyl)-5-(and-6)-Carboxyfluorescein, acetoxymethyl ester; A.G. Scientific, Göttingen, Germany) to measure pH, or OGB1-AM (Oregon Green BAPTA 1-acetoxymethyl ester; Invitrogen, Schwerte, Germany) to measure [Ca^2+^]_i_. The first two of these were used for wide-field ratiometric imaging as has been described previously [33,34,35], and were excited alternatingly by 340/380 nm and 440/490 nm, respectively. OGB1-AM was excited at 488 nm. Excitation light was produced by a PolychromeV monochromator (Thermo Fisher Scientific, Eindhoven, Netherlands). Imaging was performed using an upright microscope (Nikon Eclipse FN-1, Nikon, Düsseldorf, Germany) equipped with a Fluor 40×/0.8 W water immersion objective (Nikon). Emission was collected above 420 nm for SBFI-AM, above 505 nm for OGB1-AM, and between 511 nm and 563 nm for BCECF-AM. SR101 was excited at 575 nm and its emission collected above 590.

Images were generated with an ORCA FLASH 4.0LT camera (Hamamatsu Photonics Deutschland GmbH, Herrsching, Germany) controlled by the imaging software (Nikon NIS-Elements AR v4.5, Nikon, Düsseldorf, Germany). Emission/emission ratio was subsequently analyzed for each region of interest (ROI) using OriginPro 9.0 (OriginLab Corporation, Northampton, MA, USA) software. Changes in [Na^+^]_i_ were converted to mM on the basis of an in situ calibration performed as reported previously [33,34].

### 2.3. Determination of Baseline [Na^+^]_i_ and Electrophysiology

Baseline [Na^+^]_i_ was determined using a procedure as described in detail earlier [36]. To this end, cells were first loaded with SBFI-AM and the fluorescence ratio measured in cell bodies as described above. Subsequently, whole-cell patch-clamp recordings were performed on SR101-positive astrocytes employing an intracellular saline composed of (in mM): 114 (or 109) KMeSO_3_, 32 KCl, 10 HEPES, 10 (or 15) NaCl, 4 Mg-ATP, and 0.4 Na_3_GTP, pH adjusted to 7.3, to which 0.5 mM of the membrane-impermeable form of SBFI was added. Pipettes (2–3 MΩ) were pulled out from borosilicate glass capillaries (Hilgenberg, Malsfeld, Germany) using a vertical puller (PP-830, Narishige, Tokyo, Japan). Cells were held in voltage-clamp mode (holding potential −90 mV) using an EPC10 amplifier and PatchMaster software (HEKA Elektronik, Lambrecht, Germany).

### 2.4. Data Analysis and Statistics

Unless otherwise specified, data are illustrated in box plots indicating median (middle line), mean (red square), interquartile range (box), and standard deviation (whiskers). In addition, all individual data points are shown underneath the box plots.

Unless otherwise stated, each series of experiments was performed on at least four different animals; “n” represents the number of cells analyzed, “N” the number of individual experiments/slice preparations. Data were statistically analyzed by *t*-tests. *p*-values below 0.01 were considered to indicate a significant difference. The following symbols are used to illustrate the results of statistical tests in the figures: ** 0.001 ≤ *p* < 0.01; *** *p* < 0.001.

## 3. Results

### 3.1. Neonatal Cells Show Spontaneous Na^+^ Fluctuations

To monitor Na^+^ in astrocytes and neurons during early postnatal development, acutely isolated tissue slices of the neonatal mouse hippocampus (P2–4) were stained with SBFI-AM, a ratiometric Na^+^ indicator. Under control conditions, during perfusion with standard ACSF, we detected spontaneous fluctuations of somatic Na^+^ in a subset of SR101-positive astrocytes in the CA1 stratum radiatum and in CA1 pyramidal neurons (Figure 1(A1,A2)). Within the observation period of 60 min (standard imaging frequency 0.2 Hz), fluctuations were present in 44% of recorded astrocytes (*n* = 51/116 cells; Table 2) and in 20% of recorded neurons (*n* = 49/234 cells; Table 2, Figure 1C). The frequency of these fluctuations was very low, with the active astrocytes averaging at 1.8 ± 1.1 fluctuations/hour and the active neurons exhibiting 1.5 ± 0.9 fluctuations/hour. Notably, apparent synchronicity of fluctuations between neurons and astrocytes or within each cell group was observed only rarely and only in small subsets of responding cells (see arrowheads in Figure 1(A2,B2)).

To test if fluctuations are restricted to the neonate hippocampus, we performed recordings in acute tissue slices of neonate neocortex (layer II/III). In the neocortex, similar Na^+^ fluctuations were observed within both cell types (Figure 1B); the percentage of active cells was also comparable (39% astrocytes, *n* = 29/74; and 17% of neurons, *n* = 16/97, Figure 1C).

In order to further investigate the properties and origin of the spontaneous Na^+^ transients, all following experiments and analysis were performed in the hippocampus CA1 region. The recorded fluctuations were long-lasting (Figure 2), with an average duration of 8.7 ± 3.4 min in astrocytes (*n* = 91 fluctuations; Table 2) and 9.2 ± 4.2 min in neurons (*n* = 116 fluctuations; Table 2). The mean amplitude for fluctuations was 2.1 ± 1.3 mM and 2.2 ± 1.2 mM for astrocytes and neurons respectively (Table 2). However, it must be noted there was a very high level of variation present within the properties (amplitude, and time course) of individual fluctuations for both cell types—as is illustrated in Figure 2.

### 3.2. Fluctuations Are Developmentally Regulated

To study the developmental profile of spontaneous Na^+^ fluctuations, experiments were performed at different stages of postnatal development. Four age groups were compared for overall activity rates and for the individual fluctuation properties. These groups contained age ranges as follows: P2–4 (neonatal, example shown in Figure 3(A1)), P6–8, P10–12, and P14–32 (juvenile, example shown in Figure 3(A2)).

Activity was reduced with ongoing postnatal development in both cell groups, dropping in astrocytes from 44% (*n* = 116) in neonatal animals to 32% in P6–8 (*n* = 134) to 11.7% in P10–12 (*n* = 77) and finally to 7.9% in juvenile animals (*n* = 126; *p* = 0.001) (Figure 3(B1)). While this reduction was fairly linear in astrocytes, neuronal transients remained high in the earliest two age groups, starting at 20.1% in neonates (*n* = 234) and staying at 18.7% in the P6–8 group (*n* = 209). However, after this they had a strong reduction to 3.7% (*n* = 134; *p* = 0.028) in the P10–12 group, and stayed low at 7.4% (*n* = 242; *p* = 0.001) in the juvenile group compared to neonates (Figure 3(B1)).

The frequency of fluctuation occurrence also trended to decrease with development, with the groups showing a mean frequency (signals/hour) of 1.8 ± 1.1, 1.7 ± 0.8, 1.4 ± 0.7, and 1.1 ± 0.3 for astrocytes and 1.5 ± 0.9, 2.6 ± 1.6, 1.0 ± 0, and 1.1 ± 0.4 for neurons (groups shown in order of ascending age). Notably, while the percentage of active cells was reduced with the progression of development, the properties of the fluctuation (duration and amplitude) themselves were not altered across the age groups for either cell type (Figure 3(B2)).

### 3.3. Fluctuations Are Not Causally Linked to Ca^2+^ Signalling

Neuronal Ca^2+^ based early network oscillations (ENOs) are strongly dampened at room temperature as compared to near physiological temperatures [37]. The opposite is true for astrocytic Ca^2+^ oscillations, which can occur at a lower frequency and have shorter durations when experimental temperatures are raised to physiological conditions [38].

We therefore investigated the effect of increasing the ACSF temperature from room temperature (20–22 °C) to 33–35 °C on spontaneous Na^+^ fluctuations in neonates (P2–4). The increase in temperature had no effect on either the duration, amplitude or frequency of the fluctuations in astrocytes (cell *n* = 86, fluctuation *n* = 73), or neurons (cell *n* = 43, fluctuation *n* = 38, Figure 4A). While this does not rule out a shared origin for both spontaneous Ca^2+^ oscillations and Na^+^ fluctuations, it does suggest that the signal types are not strictly mechanistically linked.

In order to further study a possible relation between spontaneous Na^+^ fluctuations and intracellular Ca^2+^-signaling, we performed measurements with the Ca^2+^ indicator dye OGB1-AM. Under control conditions, vivid spontaneous activity, encompassing essentially the entire populations of neurons and astrocytes was observed (100% active *n* = 44/44 astrocytes; and 98% active *n* = 78/80 neurons) (Figure 4B, left column). Slices were then perfused with a nominally Ca^2+^-free saline that in addition contained 500 µM BAPTA-AM and 1 mM EGTA to chelate intra- and extracellular Ca^2+^. Under these conditions, Ca^2+^-signals were nearly completely absent (5% *n* = 2/39 astrocytes; and 3% *n* = 1/35 neurons) (Figure 4B, right column). When measuring intracellular Na^+^ however, a large increase in spontaneous fluctuation amplitude (*p* < 0.001) and duration (*p* < 0.001) in astrocytes was observed (68% *n* = 48/71) (Figure 4C) in the chelated Ca^2+^ condition. Neurons under these conditions displayed repetitive, synchronous increases in [Na^+^]_i_ (cell *n* = 52) (Figure 4D).

Taken together, these result strongly suggest that spontaneous Na^+^ fluctuations in astrocytes and neurons of the neonate brain are not causally linked to spontaneous Ca^2+^-signals detected in this age group. They also hint at a correlation between neuronal activity and astrocytic fluctuations.

### 3.4. Origin of Neuronal Na^+^ Fluctuations

The mechanism behind the generation of the developmentally regulated Na^+^ fluctuations was further studied using a broad pharmacological approach. To this end, we bath applied various blockers for ion channels, transmitter receptors, and transporters reported to be expressed by cells in this early developmental stage (summary of specific blockers used, their solvents and final concentrations can be found in Table 1). Experimental data from this pharmacological approach are illustrated in Figure 5 and detailed in Table 2.

Initially, tetrodotoxin (TTX) was introduced in order to block the activation of voltage gated Na^+^ channels. Neuronal fluctuations under these conditions had strongly reduced amplitudes (0.7 ± 0.3 mM; *p* = 0.002), eluding to the fact that they are largely dependent on the generation of action potentials. We next looked at several different signaling pathways which could be activated by action potentials during this developmental phase. Neuronal Na^+^ fluctuations and amplitudes were significantly reduced by an antagonist for GABA_A_ receptors (bicuculline; 0.7 ± 0.4 mM, *p* < 0.001), while blocking GABA_B_ receptors by CGP, had no effect. To further address the involvement of GABA in neuronal Na^+^ fluctuations, we applied nipecotic acid (NPA), a competitive inhibitor of GABA transporters (GATs), which resulted in a significant reduction (1.0 ± 0.3 mM; *p* < 0.001). The same result was obtained when introducing SNAP, a blocker for the GABA-transporter GAT3 (mainly expressed by astrocytes) together with NNC, which blocks GAT1 (that is mainly expressed by neurons) [19] (1.1 ± 0.2 mM; *p* = 0.003).

In contrast to this, pharmacological inhibition of ionotropic and metabotropic glutamate receptors and of glutamate transporters (as listed in Table 2), did not significantly influence neuronal Na^+^ fluctuations (Figure 5). The same was true for different other pathways tested, including cholinergic and purinergic receptors as well as various secondary-active, Na^+^-dependent transporters. Interestingly, inhibition of α1-noradrenergic receptors with prazosin dampened Na^+^ fluctuations in neurons (1.3 ± 0.4 mM, *p* = 0.002), while blocking ß2-receptors was without effect (Table 2, Figure 5).

These data strongly suggest that neuronal Na^+^ fluctuations in neonate hippocampus are induced by suprathreshold activity, involve the action of GABA on GABA_A_ receptors and an activation of α1-noradrenergic receptors, whereas glutamate-related pathways apparently do not play a role.

### 3.5. Pharmacology of Astrocytic Na^+^ Fluctuations

Similar to what was observed in neurons (see above), the amplitudes of astrocyte Na^+^ fluctuations were significantly reduced in the presence of TTX compared to control conditions (1.0 ± 0.3 mM; *p* < 0.001) (see Table 2 and Figure 5). In contrast to neurons, blocking of GABA_A_ receptors had no effect on astrocytic fluctuations. However, nipecotic acid resulted in a significant reduction in the amplitudes of astrocyte Na^+^ fluctuations (1.3 ± 0.6 mM; *p* = 0.003). Surprisingly, after blocking GABA transporters with SNAP and NNC, which had dampened neuronal fluctuations, we observed an increase in astrocyte amplitudes (2.9 ± 1.8 mM; *p* < 0.001).

In order to further explore the pathways for cellular generation of neonate astrocyte Na^+^ fluctuations, we first determined the baseline Na^+^ concentration in astrocyte somata of two different age groups, following a procedure as previously reported for neurons [36]. Experiments revealed a comparable baseline Na^+^ concentration of astrocytes in neonatal animals (P2–4) (12.2 ± 2.6 mM; *n* = 15) and in juvenile animals (P14–16) (11.5 ± 3.0 mM; *n* = 8). The latter values are consistent with earlier reports [27]. Next, we pharmacologically inhibited a number of ion channels, transmitter receptors and transporters, which might be linked to their generation as listed in Table 2 and depicted in Figure 5.

Blocking of neither glutamate receptors nor excitatory amino acid transporters had any effect on the fluctuations in astrocytes. We then tested the possible involvement of glycinergic transporters and cholinergic receptors. However, application of antagonists for these components also had no effect on the astrocytic fluctuations observed here. Moreover, antagonists for different purinergic receptors (P2X, P2Y, and P1 receptors) were applied, but did not alter the observed Na^+^ fluctuations in astrocytes. Furthermore, we analyzed the involvement of adrenergic signaling, but astrocyte fluctuations were not significantly altered neither by the α1 receptor antagonist prazosin (which dampened neuronal fluctuations), not by the non-selective ß-antagonist propranolol.

Next, a number of antagonists for Na^+^ dependent transporters were implemented, including antagonists for the NCX (sodium/calcium exchanger), NKCC1 (sodium/potassium/chloride co-transporter 1) and NHE (sodium/proton exchanger). The blocking of these transporters also showed no significant effect on either fluctuation properties or prevalence (Figure 5).

Finally, we bath applied lanthanum (La^3+^) which acts as a hemichannel antagonist, as well as blocking stretch-activated TRP channels and leak channels. However, this had no impact on Na^+^ fluctuation prevalence or amplitude in astrocytes (*n* = 84, *N* = 4) nor in neurons (*n* = 44, *N* = 4) (Figure 5). Albeit weak as compared to more mature tissue, there is still a well detectable degree of gap junctional coupling between astrocytes in the first postnatal week [39]. In order to test for an involvement of gap junctions, we also used mice deficient for Cx30 and 43 (experiments performed in three slices from 2 animals). The Na^+^ fluctuations in these were also found to be unaltered in astrocytes (*n* = 46) as well as in neurons (*n* = 77) of Cx-deficient mice in comparison to the respective controls (20 animals; astrocytes: *n* = 548, *N* = 38; neurons: *n* = 460, *N* = 14) (Figure 5, Table 2).

In summary, these results demonstrate that Na^+^ fluctuations in neonate astrocytes are strongly dependent on the generation of action potentials and significantly affected upon increasing the concentration of GABA in the extracellular space. Moreover, our data suggest that they are apparently not primarily mediated by activation of GABA_A_ receptors on astrocytes. Nipecotic acid, a competitive inhibitor of GABA uptake, reduced Na^+^ fluctuations to a similar degree as TTX. This effect was, however, seemingly not directly related to its blocking of GABA transporters, because application of SNAP together with NNC, which target GAT1 and GAT3, had the opposite effect. None of the many other pathways tested significantly reduced astrocyte Na^+^ fluctuations, so that their genesis remained unclear.

### 3.6. Astrocyte Fluctuations Are Not Bound to pH Regulation

Na^+^ homeostasis is bound to the regulation of pH via sodium-dependent transporters such as the NHE and the NBC (sodium-bicarbonate co-transporter) [40]. In order to determine whether the spontaneous astrocytic Na^+^ transients could be linked to changes in pH we measured [Na^+^]_i_ in astrocytes for 20 min, before washing in a HEPES-buffered ACSF. The removal of CO_2_ and HCO_3_- from the solution has been shown to reverse the NBC to transiently operate in its ‘outward’ mode, exporting Na^+^. After a few minutes, HCO_3_- is depleted from the cells, and NBC activity greatly reduced [41,42]. Evidence for this was also seen here, as switching to HEPES-buffered ACSF induced a fast, but transient drop in [Na^+^]_i_ when the solution was introduced (*n* = 62 astrocytes, Figure 6A).

Spontaneous astrocyte Na^+^ fluctuations were seen at similar levels both before and after the removal of CO_2_ and HCO_3_- (in 45% and 47% of measured astrocytes for each condition respectively). Interestingly, the fluctuations measured in HEPES-buffered ACSF were significantly longer (*p* = 0.002) and had a higher amplitude (*p* = 0.012) than those measured in standard ACSF (Figure 6B). This observation makes it unlikely that the NBC—which is likely reversed during this period—is a major driving factor in their generation.

To further address a possible involvement of pH in the spontaneous astrocyte Na^+^ fluctuations, we performed measurements using the pH-sensitive indicator dye BCECF-AM. As illustrated in Figure 6C, 60 min measurements revealed no visible fluctuations in astrocyte pH (*n* = 97), further supporting that the spontaneous Na^+^ fluctuations are not tied to pH regulation. 

## 4. Discussion

The present study demonstrates for the first time that astrocytes and neurons in neonatal mouse hippocampus and cortex undergo slow, long-lasting fluctuations in their intracellular Na^+^ concentration, which gradually decline during the first two weeks of postnatal development. Neuronal Na^+^ fluctuations in neonates are dampened upon inhibition of action potentials and seemingly related to the activation of GABA_A_ receptors. While Na^+^ fluctuations in astrocytes are strongly dependent on the generation of action potentials as well, activation of GABA_A_ did not play a role. Moreover, none of the many other pathways addressed was able to unambiguously reduce the astrocyte Na^+^ fluctuations.

### 4.1. Neuronal Na^+^ Fluctuations

As mentioned above, spontaneous Na^+^ transients have never been reported before in any cell group—therefore the first but by no means least significant result is that these were indeed recorded in the neonatal mouse hippocampus and cortex. Significant reductions in neuronal fluctuation amplitudes were seen after the application of TTX and bicuculline, linking the changes in [Na^+^]_i_ to action potential firing and the activation of GABA_A_ receptors respectively.

This suggests that GABA (likely released by interneurons) activates GABA_A_ receptors on pyramidal cells causing their depolarization, which then drives the generation of neuronal Na^+^ fluctuations, presumably via the opening of TTX-sensitive voltage gated Na^+^ channels. Additionally, the blocking of GABA uptake by both NPA and SNAP/NNC dampened neuronal fluctuation amplitudes, an effect that could be explained by reduced neuronal activity caused by the build-up of extracellular GABA. Notably, spontaneous Na^+^ fluctuations were primarily present during the first week after birth and were strongly diminished thereafter. The developmental regulation of the fluctuations along with their apparent mechanism, is similar to that which has been demonstrated to be responsible for GDPs and ENOs [37,43]. In these models the excitatory action of GABA is accounted for by high early expression of NKCC1 (sodium potassium chloride co-transporter) compared to KCC2 (potassium chloride co-transporter) which does not reach adult levels of expression until the third postnatal week [44]. This results in elevated intracellular Cl^−^ [45], reversing the Cl^−^-gradient and leading to an efflux upon GABA_A_ opening—thus depolarizing the neuron [44,46,47]. In vivo experiments have recently called the principle of excitatory GABA back into question [48,49]. Notwithstanding, our experiments clearly link GABA_A_ receptor activation and neuronal action potential generation to the slow Na^+^ fluctuations observed in the neonate brain.

Although there are similarities between the previously described phenomena and Na^+^ fluctuations found here, the link between them is not absolute. Along with the extended duration of each fluctuation, there appears to be very little synchronicity between neurons—a feature prominent among neurons for both ENOs and GDPs [7,50]. Additionally, it has been well established that neuronal Ca^2+^ based early network oscillations are almost completely blocked at room temperature [37], and the fluctuations here were unaffected in either propensity or property by alterations in experimental temperature. While these results do not rule out a shared origin for both spontaneous Ca^2+^ oscillations and Na^+^ fluctuations, it does suggest that the signal types are not directly mechanistically linked e.g., via the Na^+^/Ca^2+^ exchanger (NCX). This was again confirmed pharmacologically as the application of the NCX antagonist SEA0400 had no significant impact on fluctuation properties. Surprisingly, the pharmacological evidence shows that Na^+^ fluctuation properties remained unchanged in the presence of bumetanide—a blocker for the NKCC1 which has been shown to reduce GDP frequency in neurons [51]. NMDA signaling too—made possible by the release of the Mg^2+^ block by GABA-induced depolarization [6]—has been implicated as assisting GDP propagation [5]. However, antagonists for this, along with other glutamatergic signaling components had no effect on the Na^+^ fluctuations in this study.

Other signaling pathways were examined pharmacologically and were found to have no detectable influence on fluctuation properties, with the exception of the noradrenaline α1 receptor blocker prazosin which reduced fluctuation amplitude. While expression of α1 is low at early neonatal stages [52], activation has been shown to increase interneuron GABAergic output in the hippocampus [53] and may be upregulating activity even at this early stage.

These results suggest that the ultraslow Na^+^ fluctuations in pyramidal cells are dependent on action potential firing, GABA release by interneurons and subsequent depolarization via GABA_A_ receptors. Although this is a mechanism shared with GDPs, they appear not to be directly connected. GDPs generally constitute trains of neuronal bursts that last for hundreds of milliseconds and occur at a rate of around 0.1/s [7]. However, other patterns of electrical activity are also present in the developing brain. Along with shorter patterns like spindle bursts and gamma oscillations, long oscillations which last several minutes and have a frequency of around 3/h have been recorded (but their properties and origin not further investigated) [54]. The ultra-slow fluctuations seen in neuronal [Na^+^]_i_ here may therefore represent a gradual accumulation and decline, caused by variations in the frequency of bursting within individual cells or populations, or by energy substrate availability.

### 4.2. Astrocytic Na^+^ Fluctuations

Intriguingly, astrocytes of both the hippocampus and cortex were twice as likely to show [Na^+^]_i_ fluctuations as neurons. However, the properties of individual fluctuations did not differ significantly from their neuronal counterparts: they were unusually long (duration ~8 min) and had a very low occurrence rate (~2/h).

Unlike neuronal ENOs, synchronous, spontaneous Ca^2+^ signaling as detected in astrocyte somata first appears across several brain regions in early development and remains present (although at much lower levels) into adulthood [23,55]. While the trigger for these astrocytic Ca^2+^ signals often remained unclear, several studies have demonstrated an independence from neuronal activity and linked these signals to release of Ca^2+^ from intracellular stores [56,57,58]. However, a modulatory role for neurotransmitters (especially glutamate) has been reported [23,57]. In addition to somatic Ca^2+^ elevations, recent imaging studies uncovered the presence of highly localized, rapid Ca^2+^ signals in astrocyte processes that are apparently independent from IP_3_-mediated Ca^2+^ release [21,59,60,61,62]. Notably, while these localized Ca^2+^-signals in some cases do invade the soma, both the occurrence and temporal properties of somatic Ca^2+^-signals reported earlier are still very much different from the Na^+^ fluctuations observed here.

Aside from the obvious discrepancy in duration, frequency and synchronicity, several of the results here further strengthen the apparent independence of Na^+^ fluctuations from astrocytic Ca^2+^ signals. While the chelation of intra and extracellular Ca^2+^ almost completely eradicated spontaneous Ca^2+^ activity, it did not reduce astrocytic Na^+^ fluctuations. Indeed, Ca^2+^ chelation increased Na^+^ fluctuation amplitudes, and produced repetitive, synchronous Na^+^ transients in neurons, an effect which can be explained by a likely depolarization of neurons under these conditions [63,64].

The increase in astrocyte activity seen here along with the reduction of fluctuations by TTX suggests a strong link between astrocytic fluctuations and neuronal action potential firing. This is again confirmed by the increase in astrocyte fluctuation amplitude after the removal of CO_2_ and HCO_3_– from the ACSF, which has been shown to alkalinize neurons [42], in turn increasing excitability [40,65].

The results in the previous section show that the primary drive for neuronal Na^+^ is GABAergic signaling. While this can be significantly reduced by the inhibition of GABA_A_ receptors, doing so had no effect on astrocytic fluctuations. Inhibition of GABA uptake by NPA on the other hand resulted in a significant reduction in the amplitudes of astrocyte Na^+^ fluctuations. In contrast, application of SNAP and NNC, which had dampened neuronal fluctuations, increased astrocyte fluctuation amplitudes. As both of these blockers target GABA transporters, this result may first seem counterintuitive, however, it may be related to their different modes of function. Both NPA and SNAP/NNC prevent the reuptake of GABA into cells- thereby resulting in a buildup of GABA in the extracellular space [66,67]. The opposite outcome of the two approaches, however, suggests that this buildup is probably not directly responsible for the observed actions.

Notably, substances were applied for rather long periods, that is for 15 min prior to and for 60 min during the measurements. A likely scenario is, therefore, that blocking GABA uptake resulted in a significant depletion of interneurons’ GABA stores [68], thereby reducing its release in response to action potentials. NPA does this by acting as a competitive agonist, and is taken up into cells via GABA transporters [69]. As NPA has been reported to have a half-life of several hours inside cells [70], there would likely be considerable build-up of the acid inside cells. This acidification would reduce excitability [40,65], and thereby generally reduce interneuron activity (and release of any other transmitter or substance related to this). On the other hand, SNAP/NNC are not taken up directly [71] and also do not cause an intracellular acidification. While GABA builds up in the extracellular space, interneurons may still continue to fire action potentials—and may continue to release transmitters or substances besides GABA.

In addition, while astrocyte Na^+^ fluctuations do seem to depend on neuronal activity, the differences in reaction to bicuculline and SNAP/NNC point to an independence from pyramidal cell firing. Interneurons have been widely reported to display co-transmission, especially in developing tissue [72,73]. Together, these findings suggest that interneurons could be co-releasing GABA (driving pyramidal activity) along with another transmitter or substance which interacts with astrocytes.

Transmitters reported to be co-released in this manner include glutamate, acetylcholine, and glycine [74,75]. Components from these signaling pathways were inhibited, but these experiments showed no significant difference compared to control experiments and these pathways were thus excluded from playing a role in the astrocytic fluctuations. The lack of input from glutamatergic signaling in particular was somewhat unexpected, as glutamate has been shown to elicit Na^+^ transients in neonatal hippocampal astrocytes, mainly via EAAT based uptake [27,28]. Purinergic and adrenergic signaling pathways were also investigated, along with other Na^+^ dependent membrane transporters including the NKCC1, NHE, NBC, and NCX. However, blocking of these components did not significantly impact astrocytic fluctuations and are therefore unlikely to be involved in their generation or propagation.

There are still a number of other pathways or signals which could be responsible for the changes in [Na^+^]_i_ seen here. A possible source for Na^+^ influx might e.g., be mechanosensitive channels, and more specifically Piezo1, which was recently identified to be expressed in astrocytes [76,77]. Moreover, neuropeptides, growth factors, dopaminergic signaling, and nitric oxide are just some of the factors which have been shown to influence development and cellular physiology during development and might be linked to Na^+^ influx. Additionally, because of the low expression levels of Kir4.1 channels in newborn animals, fluctuations in the membrane potential could contribute to the slow Na^+^ fluctuations [15]. Alternatively, it is possible that what are perceived here as elevations are in fact due to the down-regulation of Na^+^ extrusion, e.g., due to a transient inhibition of the NKA activity via cellular signaling pathways and/or metabolism.

Another point which remains to be clarified is which interneurons are specifically responsible for neuronal and astrocytic fluctuations and whether one group is simultaneously innervating both cell types or two separate subpopulations are interacting with one cell type each. The former system could offer an explanation for the lack of synchronicity and different levels of activation seen in astrocytes and neurons.

Of note, it cannot be entirely excluded that the slow Na^+^ fluctuations are an artefact produced by preparation of brain slices, which still does not resolve the question of their origin. Testing for the presence of the fluctuations in the mouse brain in vivo, however, will be challenging as any anesthetics applied to the animal during the measurements could also have effects on signaling, as has been shown to be the case for Ca^2+^ signals (e.g., [78]). Neonatally restricted electrical activity first measured in acute tissue slices has now also been shown both in vivo (mouse) and in utero human EEG recordings [54,79]. The time frame for these patterns is very similar to the Na^+^ fluctuations seen here, suggesting that they too are not merely the result of the preparation method.

## 5. Conclusions

Taken together our results reveal for the first time that there exists spontaneous Na^+^ fluctuations in both neurons and astrocytes of the hippocampus and cortex within the early postnatal brain. They show that these fluctuations are developmentally regulated, but do not appear to be directly linked to Ca^2+^ signaling, which has previously been reported within the same age range. Both astrocytic and pyramidal cell fluctuations usually last several minutes, and are thus termed “ultraslow Na^+^ fluctuations”. Both appear to be driven by action potential activity in interneurons. In the model suggested by Figure 7, GABA released by this activity goes on to depolarize pyramidal cells via GABA_A_ receptors, which in turn depolarizes the cells and opens voltage gated Na^+^ channels, leading to bursts of action potentials underlying the neuronal Na^+^ fluctuations. Astrocytic Na^+^ fluctuations also appear to rely on interneuron firing, but their exact signaling mechanism excludes all indicated signaling pathways and remains to be determined.

## Figures and Tables

**Figure 1 cells-09-00102-f001:**
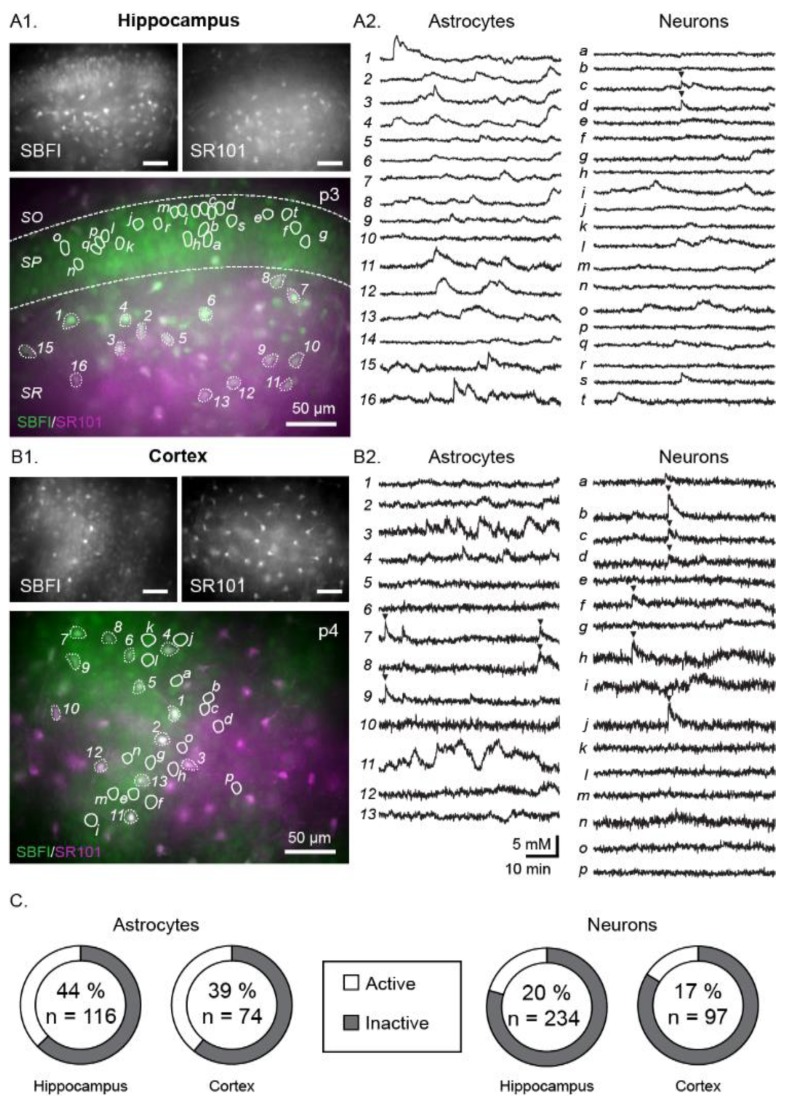
Example measurements showing spontaneous Na^+^ fluctuations in the neonate hippocampus (**A1,A2**) and neocortex (**B1,B2**). (**A1**,**B1**) show SBFI (top left), SR101 (top right) and merge images (bottom) with all scale bars indicating 50 µm. Circled areas correspond to regions of interest (ROIs), the individual fluorescent measurement traces of which are illustrated in A2 and B2 (astrocytes on the left and numbered, neurons on the right and labelled with letters). Arrows indicate instances when cells appear to be synchronized. (**C**) Pie charts indicating the percentage of active astrocytes (left) and neurons (right) within each area (*n* = total number of cells measured). SBFI: sodium-binding benzofuran isophthalate-acetoxymethyl ester.

**Figure 2 cells-09-00102-f002:**
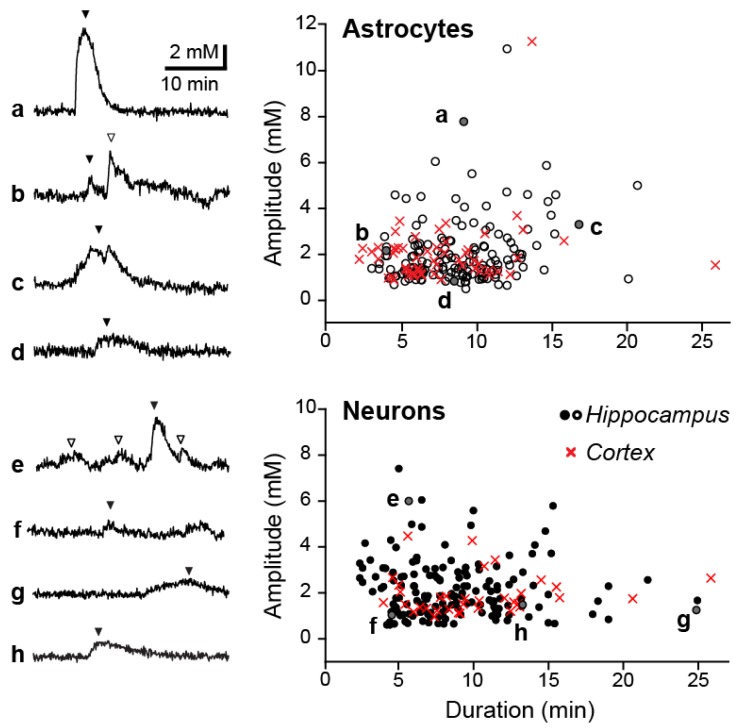
Properties of spontaneous Na^+^ fluctuations. Left: Examples of individual Na^+^ fluctuations in astrocytes (**a**–**d**) and neurons (**e**–**h**). Right: Scatter plot of all measured fluctuations in astrocytes (top) and neurons (bottom) with examples indicated by letters and shown by filled arrowheads on the left (hollow arrowheads are aditional fluctuations also analyzed).

**Figure 3 cells-09-00102-f003:**
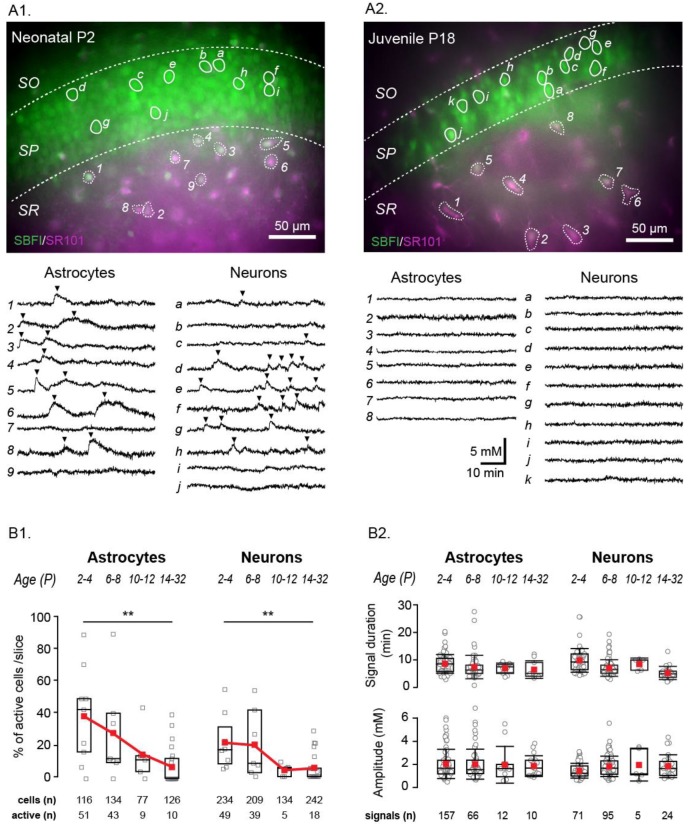
Age dependence of spontaneous Na^+^ fluctuations. Example merged staining of a P2 (**A1**) and P18 (**A2**) hippocampal slice (SBFI in green, SR101 in magenta, overlapping regions in white) with example cells encircled and their individual fluorescent measurement traces shown below. Analyzed fluctuations are indicated by black arrows. Scale bars show 50 µm. SO, SP, SR: stratum oriens, pyramidale and radiatum, respectively. (**B1**) Box plots illustrating the decline in percentage of cells showing activity per slice with increasing mouse age. (**B2**) Properties from fluctuations in different age groups as written above. Box-and-whisker plots indicating median (line), mean (red square), interquartile range (box) and standard deviation (whiskers). ** 0.001 ≤ *p* < 0.01.

**Figure 4 cells-09-00102-f004:**
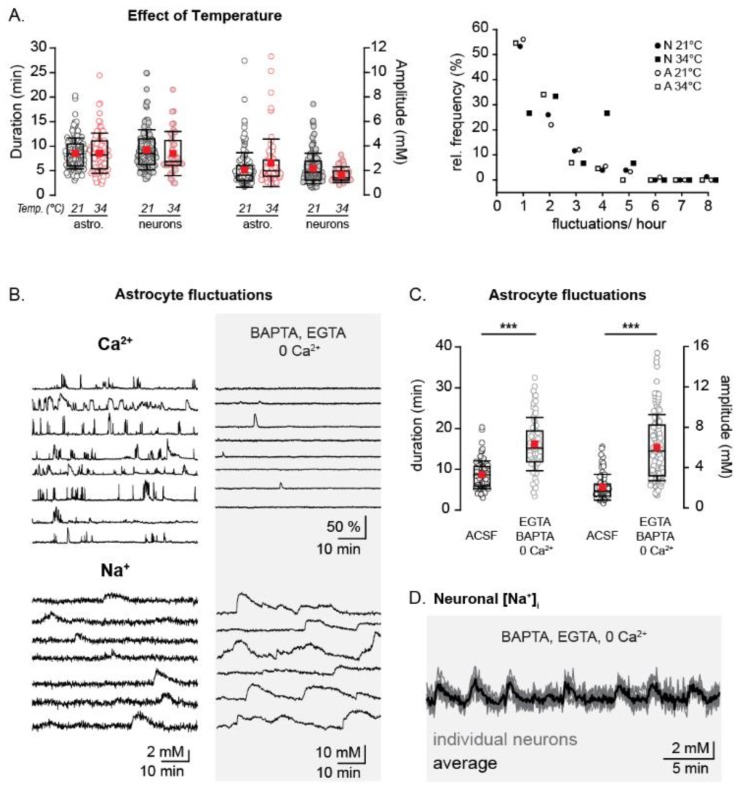
Temperature dependence and interrelation with spontaneous Ca^2+^-signaling. (**A**) Left: duration and amplitude of neonatal astrocytic and neuronal Na^+^ fluctuations at 21 °C (black) and 34 °C (red). Right: Relative frequency distribution plot of neonatal Na^+^ fluctuations at 21 °C (neurons- black circles, astrocytes- white circles) and 34 °C (neurons- black squares, astrocytes- white squares). (**B**) Example traces from individual astrocytes in Ca^2+^ imaging experiments (top) and during Na^+^ imaging (bottom), both in ACSF (left) and in the presence of a Ca^2+^ chelated solution (0 Ca^2+^, ACSF containing 500 µM BAPTA-AM and 1 mM EGTA) (right). (**C**) Box plots illustrating the increase in astrocytic Na^+^ fluctuation amplitudes after the chelation of Ca^2+^. (**D**) Traces showing Na^+^ fluctuations in several individual neurons (grey) and an averaged trace (black) in a Ca^2+^ chelated solution. *** *p* < 0.001.

**Figure 5 cells-09-00102-f005:**
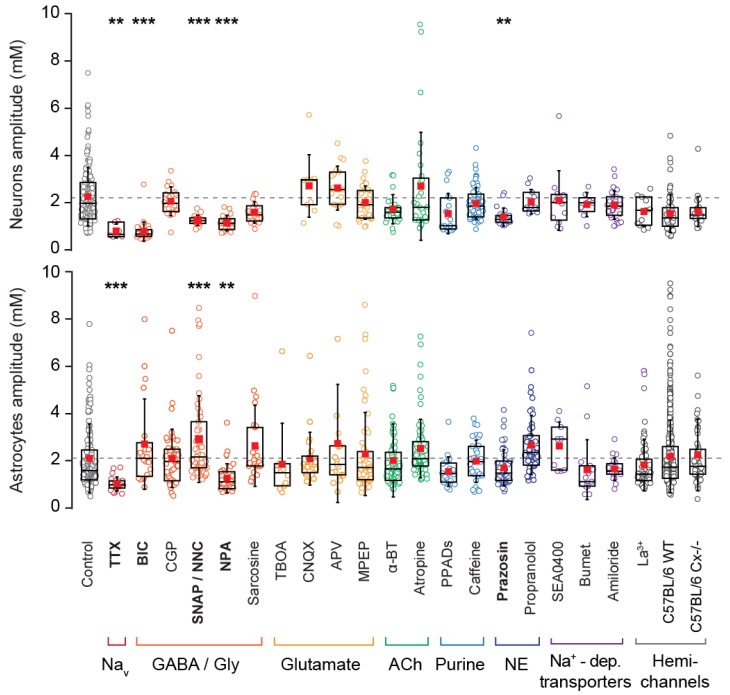
Pharmacological investigation. Comparison of fluctuation amplitudes in neurons and astrocytes in the presence of various pharmacological conditions as detailed in Table 1 and Table 2. Blockers are arranged into color groups according to their target pathway as indicated below the plots. Abbreviations, see Table 1. Bumet: bumetanide; a-BT: a-bungarotoxin. ** 0.001 ≤ *p* < 0.01; *** *p* < 0.001.

**Figure 6 cells-09-00102-f006:**
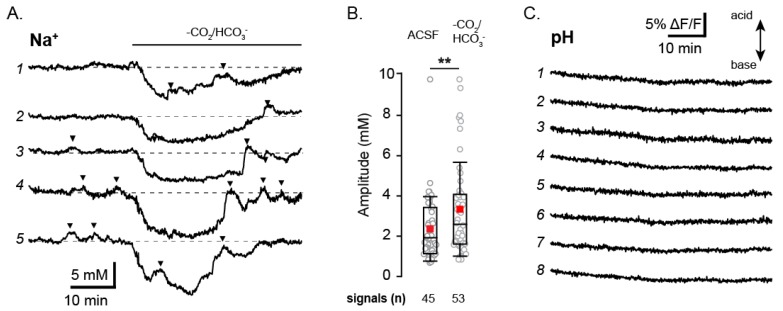
The influence of pH on astrocytic Na^+^ fluctuations. (**A**) Shows five example astrocytes traces from a single SBFI experiment wherein HEPES buffered ACSF was washed in after 20 min of baseline measurement. Analyzed fluctuations are identified by black arrows. (**B**) Box plots comparing control condition fluctuation amplitudes to those measured in the HEPES buffered ACSF. (**C**) BCECF fluorescence traces from 8 individual astrocytes in a single experiment. ** 0.001 ≤ *p* < 0.01.

**Figure 7 cells-09-00102-f007:**
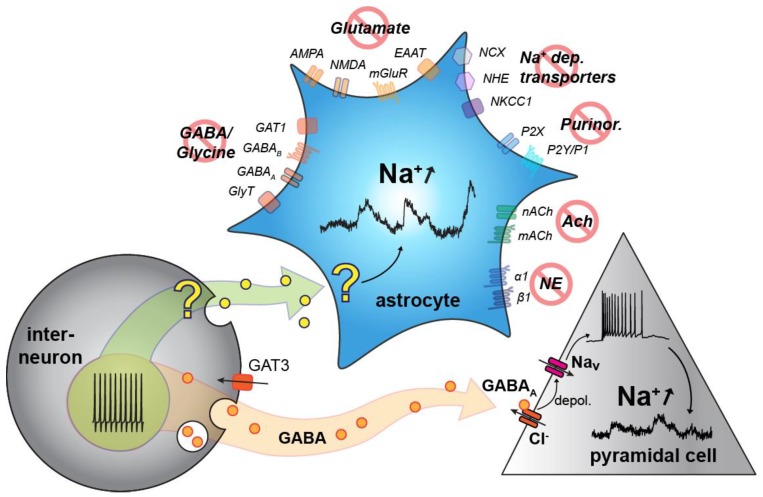
Summary of results. Interneuron firing (bottom left) stimulated release of GABA, which depolarises pyramidal cells (bottom right) via GABA_A_ activation and subsequent Cl^−^ efflux. This opens voltage gated Na^+^ channels which produces neuronal Na^+^ fluctuations. Interneuron activty also appears to be responsible for astrocytic fluctuations. These appear to have a separate trigger and mechanism, the source of which remains unclear at present. Abbreviations, see Table 1. NE: nor-epinephrine; Ach: acetylcholine; depol: depolarization; Na_v_: TTX-sensitive voltage-gated Na^+^ channel.

**Table 1 cells-09-00102-t001:** Blockers used.

Target Group	Blocker	Target	Solvent	Conc. (µM)	Manufacturer	Catalogue Number
**Glutamatergic**	MPEP	mGluR5	DMSO	25	Tocris	1212
	TFB-TBOA	EAATs	DMSO	1	Tocris	2532
	APV	NMDA	NaHCO_3_	100	Cayman Chem.	14,540
	CNQX	AMPA	DMSO	25	Cayman Chem.	14,618
**GABAergic/** **Glycinergic**	Bicuculline	GABA_A_	σ H2O	10	Sigma-Aldrich	14,343
	CGP-55845	GABA_B_	σ H2O	5	Sigma-Aldrich	SML0594
	NNC-711	GAT1	DMSO	100	Tocris	1779
	SNAP-5114	GAT2/3	DMSO	100	Sigma-Aldrich	S1069
	NPA	GAT1/2/3	σ H_2_O	100	Tocris	0236
	Sarcosine	GlyTs	σ H_2_O	500	Sigma-Aldrich	131,776
**Cholinergic**	α-Bungarotoxin	α7 nAChR	σ H_2_O	0.1	Sigma-Aldrich	T0195
	Atropine	mAChR	σ H_2_O	5	Sigma-Aldrich	A0132
**Purinergic**	PPADs	P2X/Y	σ H_2_O	20	Sigma-Aldrich	P178
	Caffeine	P1	σ H_2_O	100	Sigma-Aldrich	C0750
**Adrenergic**	Prazosin	α1 receptor	σ H_2_O	0.2	Sigma-Aldrich	P7791
	Propranolol	β receptor	σ H_2_O	10	Sigma-Aldrich	P0084
**Na^+^ dependant transporters**	TTX	Na_v_	σ H_2_O	0.5	BioTrend	BN0518
	SEA0400	NCX	DMSO	10	MCE	HY15515
	Bumetanide	NKCC1	DMSO	40	BioTrend	BG0113
**pH**	Amiloride	NHE	σ H_2_O	50	Sigma-Aldrich	BP008
**Hemichannels**	La^3+^	TRP/Hemicha.	σ H_2_O	50	Merck	203,521

Abbreviations are as follows: **MPEP** (2-Methyl-6-(phenylethynyl)pyridine), **TFB-TBOA** ((3*S*)-3-[[3-[[4-(Trifluoromethyl)benzoyl]amino]phenyl]methoxy]-l-aspartic acid), **APV** ((2*R*)-amino-5-phos-phonovaleric acid; (2*R*)-amino-5-phosphonopentanoate), **CNQX** (6-cyano-7-nitroquinoxaline-2,3-dione), **CPG-55845** ((2S)-3-[[(1*S*)-1-(3,4-dichlorophenyl)ethyl]amino-2-hydro xypropyl] (phenyl-methyl)phosphinic acid hydrochloride), **NNC-711** (1,2,5,6-tetrahydro-1-[2-[[(diphenylmethylene) amino]oxy]ethyl]-3-pyridinecarboxylic acid hydrochloride), **SNAP-5114** (1-[2-[tris(4-methoxy-phenyl)methoxy]ethyl]-(*S*)-3-piperidinecarboxylic acid), **NPA** (nipecotic acid), **PPADs** (pyridoxal-phosphate-6-azophenyl-2’,4’-disulfonic acid), **TTX** (tetrodotoxin), **SEA0400** (2-[4-[(2,5-difluoro-phenyl)methoxy]phenoxy]-5-ethoxyaniline), **mGluR5** (metabotropic glutamate receptor 5), **EAATs** (excitatory amino acid transporters), **NMDA** (*N*-methyl-d-aspartate receptors), **AMPA** (α-amino-3-hydroxy-5-methyl-4-isoxazolepropionic acid receptor), **GAT** (gamma-aminobutyric acid trans-porters), **GlyT** (glycine transporters), **nACh** (nicotinic acetylcholine receptors), **mACh** (muscarinic acetylcholine receptors), **Na_v_** (voltage gated sodium channels), **NCX** (sodium calcium exchanger), **NKCC1** (sodium potassium chloride co-transporter), **NHE** (sodium proton exchanger), **TRP** (transient receptor potential channels). Manufacturers were located as follows (third party distributers are indicated in bold): Merck KGaA, Darmstadt, Germany; Biotrend, Köln, Germany; MCE, **Hycultec**, Beutelsbach, Germany; Tocris, **Bio-Techne GmbH**, Wiesbaden Germany; A.G. Scientific, **Mobitec**, Göttingen Germany; Cayman chemical, **Biomol GmbH**, Hamburg Germany; Sigma-Aldrich Chemie GmbH, Munich, Germany.

**Table 2 cells-09-00102-t002:** Number of neurons (left) and astrocytes (right) measured, the % of these showing activity, the total number of fluctuations analyzed under each condition and the mean and standard deviation for the amplitude (mM) and duration (minutes) for analyzed fluctuations.

		Neurons									Astrocytes							
		Cells (n)	% Cells (n) Active	Fluctuation (n)	mM Mean	mM SD	*p * Values	Min Mean	Min SD		Cells (n)	% Active	Fluctuation (n)	mM Mean	mM SD	*p * Values	Min Mean	Min SD
**Control**		**234**	**20.4**	**116**	**2.2**	**1.2**		**9.2**	**4.2**		**116**	**44**	**91**	**2.1**	**1.3**		**8.7**	**3.4**
**TTX**		**94**	7.4	7	**0.7**	**0.3**	******	8.2	3.7		**39**	35.9	23	**1**	**0.3**	*******	8.7	4.1
**BIC**		**76**	19.7	32	**0.8**	**1**	*******	6.6	2.3		**22**	59.1	20	2.7	1.9	0.06	8.1	2.3
**CGP**		**101**	12.9	20	2	0.6	0.48	10.9	3.5		**49**	44.9	42	2.1	1.2	0.78	9.6	4.6
**SNAP/NNC**		**53**	22.6	17	**1.1**	**0.2**	*******	10.2	3.1		**86**	34.9	33	**3.2**	**2.0**	*******	10.8	6.2
**NPA**		**70**	15.7	20	**1**	**0.3**	*******	10.7	3.8		**43**	32.6	31	**1.3**	**0.6**	******	9.4	3.6
**Sarcosine**		**74**	20.3	20	1.5	0.5	0.02	11.3	3.7		**62**	21	25	2.6	1.7	0.06	8.5	3.5
**TFB-TBOA**											**36**	27.8	10	1.9	1.7	0.69	8.5	3.7
**CNQX**		**130**	6.2	9	2.6	1.3	0.29	9.6	4.1		**45**	37.8	25	2.1	1.1	0.86	8.9	3.1
**APV**		**156**	10.3	19	2.5	0.9	0.22	10.9	5.2		**52**	28.8	19	2.7	2.5	0.34	13.1	4.9
**MPEP**		**117**	16.2	30	1.9	0.7	0.28	9.5	3.8		**26**	46.2	53	3.9	2.2	0.36	9.1	6.2
**α-BT**		**118**	11.9	29	1.6	0.6	0.02	10.1	3.7		**43**	67.4	89	2	1.5	0.93	7.3	3.7
**Atropine**		**124**	16.1	28	2.6	1.3	0.14	8.5	2.9		**73**	49.3	74	2.5	1.2	0.02	7.7	2.9
**PPADs**		**70**	11.4	19	1.4	0.9	0.01	10.7	4.9		**57**	24.6	24	1.6	0.6	0.1	8	2.4
**Caffeine**		**142**	22.5	75	1.9	0.7	0.05	9.4	4.8		**32**	62.5	33	2	0.9	0.84	8.8	3.2
**Prazosin**		**111**	13.5	21	**1.3**	**0.4**	******	9.7	3.7		**81**	40.7	50	1.7	0.8	0.18	10.4	4.6
**Propranolol**		**123**	7.3	11	1.9	0.5	0.55	11.4	4.3		**59**	38.9	54	2.7	1.2	0.01	8.7	4.1
**SEA0400**		**85**	11.7	12	2	1.3	0.65	8.8	3.1		**18**	44.4	9	2.6	1	0.21	8	1.4
**Bumet**		**132**	6.1	8	1.8	0.5	0.44	11	3.1		**29**	37.9	15	1.6	1.3	0.28	8.5	3.5
**Amiloride**		**162**	11.1	31	1.8	0.6	0.11	8.7	2.7		**50**	28	20	1.7	0.5	0.34	8.8	3.5
**La^3+^**		**144**	5.2	8	1.8	0.6	0.43	11.9	5.3		**84**	40.5	74	2.0	1.2	0.71	8.4	3.4

**C57Bl/6 WT**		**460**	11.1	66	1.5	0.8		10.8	4.9		**548**	51.1	620	2.1	1.5		8.7	4.0
**Cx -/-**		**77**	18.2	30	1.6	0.6	0.65	11.4	5.6		**46**	60.1	58	2.2	1.5	0.77	8.7	4.3

Abbreviations, see Table 1. Bumet: bumetanide; a-BT: a-bungarotoxin. Red color indicates statistically significant difference as compared to the control condition, *p*-values indicated apply to the amplitudes, ** 0.001 ≤ *p* < 0.01; *** *p* < 0.001.

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
