# Peer review of "Spontaneous Ultraslow Na+ Fluctuations in the Neonatal Mouse Brain"

_cells, 2019, doi:10.3390/cells9010102_

Round 1

Reviewer 1 Report

This is an elegant and carefully executed study that characterized slow oscillations in intracellular Na+ levels in immature brain, separately in neuronal and astrocytic cells. 

The conclusions of this manuscript are sound, and the discussion of presented data is balanced and not overreaching.  Several potential mechanisms have been probed using pharmacological tools.  Although pharmacological agents always come with the caveat of specificity and whether they reach effective levels, the group of Dr. Rose has employed these and other pharmacological tools for many years.  Consequently, they accumulated significant experience and positive data for majority (all?) of the agents that have been employed in the present work.

I think that this manuscript will be of significant interest to specialists in the fields of bran development and neuronal and glial biology.  For this reason and based on the quality of the work this study clearly belongs to this journal.

I have two minor suggestions for increasing transparency of data presentation:

[1] Give catalogue numbers for all compounds listed in Table 1 and imaging agents (this will allow to check their listed purities and assist in data reproduction).

[2]  Present in an additional Table or tables the data on Na+ oscillations in Cx30 and Cx43 KO animals and in the presence of La3+.  This is important because through these experiments the Authors ruled out (or confirmed ruling out) contribution of gap junction connectivity and hemichannels, leak channels, mechanosensitive channels in observed Na+ oscillations.  Yet, because the numbers are small, I would allow reader to see the quality of the data.  An additional caveat of La3+ experiments is that La3+ may precipitate in phosphate-buffered solutions (I hope that the Authors took this into account).

Author Response

1) Give catalogue numbers for all compounds listed in Table 1 and imaging agents (this will allow to check their listed purities and assist in data reproduction)

Our Response: Thank you for this comment. We followed this suggestion and have now listed the CAS numbers for each component in Table 1.

 2) Present in an additional Table or tables the data on Na+ oscillations in Cx30 and Cx43 KO animals and in the presence of La3+.  This is important because through these experiments the Authors ruled out (or confirmed ruling out) contribution of gap junction connectivity and hemichannels, leak channels, mechanosensitive channels in observed Na+ oscillations.  Yet, because the numbers are small, I would allow reader to see the quality of the data.  An additional caveat of La3+ experiments is that La3+ may precipitate in phosphate-buffered solutions (I hope that the Authors took this into account).

Our Response: Thank you for this comment. We followed this suggestion and now illustrate the data from both sets of experiments (Cx-k.o. animals with their relevant controls as well as La3+) in Figure 5. Moreover, the respective results are now described in more detail in the text (page 9, lines 324-333) and were included into Table 2.

The precipitation is an interesting point- we did not do specific control experiments to check the effectiveness of La3+ as it’s use is well established, however, we also did not observe any precipitate forming in the saline containing La3+ prior to or during the experiments.

Reviewer 2 Report

Lisa et cols. found that pyramidal neurons and astrocyte in neonatal mouse hippocampus and neocortex exhibit transient fluctuations in intracellular Na+. Pyramidal cells showed relation with GABA, while glial cells seem no relation with some neurotransmitters. 

Overall, the results look attractive, however as performed in brain slices not all the physiology it is preserved.

I have some concerns:

For preparing tissue slices authors use different aCSF composition for postnatal days. They have previous experience with this solutions ? Please cite any reference. If not, the ionic differences between aCSFs could affect to the results observed (different composition used for different postnatal days). This point should be clarified.  Did you measure other ionic variations? (i.e. K and/or CL). It would be interesting if Na fluctuations affects to K or CL concentrations.  Na fluctuations in neurons seems related with GABA..have you check what happens by blocking the CL channel or some Cl transporter, i.e. KCC2? The discussion should be based on results observed.  It is intriguing that astrocytes show twice Na fluctuations compared with neurons and no relation with neurotransmitter was found. How do you explain this? It could be an artefact of the brain slices model without preserving the brain physiology? Please discuss this point.  As observed, Na fluctuations in neurons decreased with age..so how explain the authors this point related with the action potential in adulthood neurons? For the figure 7 is missing the Na channel aperture in neurons (results observed) and it is showed Cl channel open (again, did you check this?). For astrocytes seems that all transporter indicated are related with the results and it is the opposite. Please indicate question marks or difference in contrast or something indicating that is a negative results. Also, putative interaction between astrocytes and neurons are no indicated, at least indicate an arrow between them. 

Author Response

1) For preparing tissue slices authors use different aCSF composition for postnatal days. They have previous experience with this solutions? Please cite any reference. If not, the ionic differences between aCSFs could affect to the results observed (different composition used for different postnatal days). This point should be clarified.

Our Response: Thank you for this comment. The use of ACSF containing reduced CaCl2 and increased MgCl2 during the slicing of juvenile tissue (postnatal days 10-17) is a standard procedure employed in many laboratories performing physiological studies including ours (most recently in Ziemens et al., J Neurosci 2019), as it results in better tissue preservation and viability by suppressing transmitter (glutamate) release during the cutting. Because neonatal tissue (animals younger than P10) is less prone to excitotoxicity, slicing is usually done with standard concentrations of CaCl2 and MgCl2. The additional sucrose ACSF used for older animals has been established by Henneberger & Rusakov (Nature Protocols, 2012) to produce healthier slices in animals older than ~3 weeks of age. After cutting, all slices were directly transferred to standard ACSF.

These clarifications have been included in the text (page 3, lines 87-100).

2). Did you measure other ionic variations? (i.e. K and/or CL). It would be interesting if Na fluctuations affects to K or CL concentrations.
Our Response: Thank you for this comment. We have studied three different ionic species, Na+, Ca2+ and H+. We fully agree that including further ions, especially K+ and Cl-, would be very interesting as well. The latter two, however, are notoriously difficult to analyze because the available fluorescence-based sensors suffer from low resolution and signal-to-noise-ratio as well as lack of specificity. Such analysis will, therefore, require elaborate and time-consuming future studies.

3) Na fluctuations in neurons seems related with GABA. have you check what happens by blocking the CL channel or some Cl transporter, i.e. KCC2?

Our Response: Thank you for this comment. Expression of KCC2 on neurons is very low during the neonatal stage, instead the primary Cl- transporter on both astrocytes and neurons is the NKCC1 (Rivera et al., Nature 1999), which we addressed via application of bumetanide. As bumetanide has in fact been shown to suppress GABAergic neonatal hippocampal activity (Sipilä et al., J Physiol 2006; Dzhala et al., Nature Medicine 2005), the result that it had no effect on the apparently GABAergic fluctuations was indeed surprising, and we have now highlighted this unexpected result in the text (page 15 lines 414-416 and 429).

4) The discussion should be based on results observed.  It is intriguing that astrocytes show twice Na fluctuations compared with neurons and no relation with neurotransmitter was found. How do you explain this?

Our Response: Thank you for this comment. We agree that origin of the astrocytic fluctuations is an intriguing open question. However, as we state in the discussion (page 16/17, lines 515-524), there are a number of possible pathways e.g. ‘neuropeptides, growth factors, dopaminergic signaling and nitric oxide.’

The pathways and mechanisms driving neonatal development are highly complex, and poorly understood in relation to their mature counterparts. We therefore primarily chose candidates to test based on their involvement in other described early activity, and their presence in the adult CA1 region. However, this is by no means to say that the astrocytic fluctuations could not be related to some other form of transmission, which diminishes with development. Indeed, the dependence of astrocytic fluctuations on neuronal firing suggests that interneuron activity is responsible for the release of a substance aside from GABA, which specifically interacts with astrocytes.

It is possible that the astrocytes are twice as likely to have Na+ fluctuations because of their situation in the stratum radiatum in the vicinity of interneurons, however, it is also possible that innervation of neurons and astrocytes is performed by two entirely different populations of interneurons, which show different levels of activity. Alternatively, the specific mechanism involved in the generation of astrocytic fluctuations may just by more sensitive.

These questions cannot be answered until it is determined exactly what pathway is responsible for astrocytic fluctuations, but we have expanded on the discussion of the possibilities within the text as suggested (page 17, lines 525-529).

5) It could be an artefact of the brain slices model without preserving the brain physiology? Please discuss this point.

Our Response: Thank you for this comment. This is a valid point, and ideally we would like to check the presence of the fluctuations in-vivo. However, any anesthetics applied to the animal during the measurements could also have effects on signaling, as has been shown to be the case for Ca2+ signals (Thrane et al., PNAS 2012). Furthermore, neonatally restricted electrical activity first measured in acute slices has now also been shown both in-vivo mouse models and in utero human EEG recordings (Yang et al., J Neurosci 2009; Khazipov & Luhmann, TINS 2006). The time frame for these patterns is very similar to the Na+ fluctuations seen here, suggesting that they too are not merely the result of the preparation method.

To clarify these points, we have added them to the discussion as suggested (page 17, lines 530-537).

6) As observed, Na fluctuations in neurons decreased with age so how explain the authors this point related with the action potential in adulthood neurons?

Our Response: We would argue that the signals appear to be the product of trains of action potentials, triggered by GABA signaling from interneurons which depolarize the pyramidal cells. As GABAergic excitation is confined to the first postnatal week (due to expression thereafter of the KCC2 and thereby the ‘chloride switch’), it makes sense that these fluctuations would also reduce with ongoing development (as has been reported with others patterns of activity such as GDPs).  

7) For the figure 7 is missing the Na channel aperture in neurons (results observed) and it is showed Cl channel open (again, did you check this?). For astrocytes seems that all transporter indicated are related with the results and it is the opposite. Please indicate question marks or difference in contrast or something indicating that is a negative results. Also, putative interaction between astrocytes and neurons are no indicated, at least indicate an arrow between them.

Our Response: Thank you for this comment. We have revised Figure 7 as suggested for better clarification. As the pyramidal cell fluctuations can be significantly reduced without an apparent effect on astrocytic fluctuations, we would propose that there is no direct correlation between the production of these two. However, astrocytic fluctuations DO appear to depend on action potential firing, presumably then from interneurons. While we were not able to elucidate the exact mechanism in this study, the green arrow with yellow signaling molecules and question marks is intended to illustrate the unknown components of the pathway. We have further highlighted them in the figure for better visualization.

Round 2

Reviewer 2 Report

The manuscript deserves to be published on cells, authors addressed very well my concerns.